# Machine Learning Models for Weight-Bearing Activity Type Recognition Based on Accelerometry in Postmenopausal Women

**DOI:** 10.3390/s22239176

**Published:** 2022-11-25

**Authors:** Cameron J. Huggins, Rebecca Clarke, Daniel Abasolo, Erreka Gil-Rey, Jonathan H. Tobias, Kevin Deere, Sarah J. Allison

**Affiliations:** 1Centre for Biomedical Engineering, School of Mechanical Engineering Sciences, Faculty of Engineering and Physical Sciences, University of Surrey, Guildford GU2 7XH, UK; 2Faculty of Psychology and Education, University of Deusto, 20012 San Sebastián, Spain; 3Musculoskeletal Research Unit, Translational Health Sciences, Bristol Medical School, University of Bristol, Bristol BS10 5NB, UK; 4MRC Integrative Epidemiology Unit, Bristol Medical School, University of Bristol, Bristol BS8 2BN, UK; 5School of Health and Life Sciences, Teesside University, Middlesbrough TS1 3BX, UK; 6School of Bioscience and Medicine, University of Surrey, Guildford GU2 7XH, UK

**Keywords:** machine learning, signal processing, activity type recognition, accelerometry, classification

## Abstract

Hip-worn triaxial accelerometers are widely used to assess physical activity in terms of energy expenditure. Methods for classification in terms of different types of activity of relevance to the skeleton in populations at risk of osteoporosis are not currently available. This publication aims to assess the accuracy of four machine learning models on binary (standing and walking) and tertiary (standing, walking, and jogging) classification tasks in postmenopausal women. Eighty women performed a shuttle test on an indoor track, of which thirty performed the same test on an indoor treadmill. The raw accelerometer data were pre-processed, converted into eighteen different features and then combined into nine unique feature sets. The four machine learning models were evaluated using three different validation methods. Using the leave-one-out validation method, the highest average accuracy for the binary classification model, 99.61%, was produced by a k-NN Manhattan classifier using a basic statistical feature set. For the tertiary classification model, the highest average accuracy, 94.04%, was produced by a k-NN Manhattan classifier using a feature set that included all 18 features. The methods and classifiers within this study can be applied to accelerometer data to more accurately characterize weight-bearing activity which are important to skeletal health.

## 1. Introduction

Osteoporosis is a major public health problem that most often affects postmenopausal women. One in three of all postmenopausal women aged 50 years or older will have an osteoporotic fracture. These fractures often result in significant pain, loss of independence, and increased morbidity and mortality [1,2]. Physical inactivity is a modifiable risk factor for osteoporosis, and regular physical activity positively affects bone health [3,4]. Physical Activity (PA) guidelines recommend the accumulation of at least 150 min of moderate PA each week to benefit general health and fitness [5], but there is a lack of specific PA recommendations for reducing the risk of poor bone health. This is partly because there is a lack of research that has accurately assessed weight-bearing physical activity relevant to the skeleton.

Accelerometers worn at the waist are the most widely used method for objective assessment of habitual weight-bearing PA in population-based studies aimed at characterising relationships with different health related outcomes [6,7]. This device is also used to provide objective outcome measures for interventions intended to increase weight-bearing PA [8]. However, there are important limitations in how accelerometry data are processed and analysed, which have major implications in the conclusions drawn from studies using this technology. Activity is generally defined in terms of duration above thresholds, based on proprietary counts per minute (cpm), denoting sedentary, light, moderate, and vigorous intensity PA [9]. The application of cut-point methods to accelerometer data in this way is associated with large misclassification errors [10]. In addition, translation of findings from epidemiological research into public health recommendations would be considerably helped if results were based on directly observed relationships with specific types of activity, such as walking or running/jogging. There is now a shift in emphasis away from cut-points, but analytical challenges remain in terms of how to characterise specific activities from raw accelerometry data [11]. A further limitation is that count-based thresholds based on measured energy expenditure may be relevant for studying relationships between PA and obesity and related outcomes such as cardiovascular disease and type II diabetes, but different measures of PA intensity may be needed when studying other health outcomes such as bone mineral density.

Several attempts have been made to classify walking and other types of weight-bearing PA based outputs of accelerometers attached to the centre of mass. One approach is to detect sit to stand transitions by combining features from accelerometers and gyroscopes [12]. Subsequent studies have utilised machine learning techniques for activity class prediction from raw accelerometer signals [13], whereby algorithms are developed using a training set comprising labelled data obtained from individuals performing a pre-determined set of activities. Using this approach studies developing machine learning models have commonly used support vector machine (SVM), random forest and artificial neural networks to classify activities with relatively high accuracies from wrist-worn accelerometers [14,15] or thigh and back worn accelerometers in children and adults [16]. However, equivalent classifiers are yet to be developed for activity recognition in waist worn accelerometer outputs from postmenopausal women, in whom movement patterns related to specific activities differ substantially compared to younger individuals.

The purpose of this study was to evaluate four machine learning models (k-Nearest Neighbours (NN) Manhattan, k-NN Euclidian, Decision Tree (DT) and Support Vector Machine (SVM)) on a binary movement classification (standing and walking) and a tertiary movement classification (standing, walking, and jogging) using accelerometry data from postmenopausal women. These models were chosen as they have been used by previous authors in the field of human movement classification. They are relatively simple to implement, robust and have previously provided accurate results for physical activity classification on different activity types compared to this study [17,18]. In regard to data pre-processing, combinations of raw data and feature sets have been investigated as the input to machine learning modes and have previously provided high accuracy results [19]. However, an optimal set of features for activity-type recognition have not been found and may also differ depending on the model used. Therefore, it is hypothesised that, by using novel combinations of features extracted from the data as inputs to the machine learning models, high levels of accuracy can be achieved for binary and tertiary movement classification.

## 2. Materials and Methods

### 2.1. Participamts

Eighty healthy postmenopausal women from Pamplona, Spain (mean age 58.4 ± 5.2 years; mean body mass 68.4 ± 9.6 kg; mean height 158.0 ± 6.7 cm) were recruited via advertisements placed at health medical centres to perform a submaximal incremental shuttle test on an indoor track, of which 30 also completed an incremental shuttle test on a treadmill, whilst wearing a hip-worn triaxial accelerometer [20,21]. A detailed description of the inclusion and exclusion criteria has been published previously by Gil-Rey et al. [20]. The local hospital’s ethical committee approved the study (Pyto2011/71) and written informed consent was obtained from all participants before any study procedures were undertaken. The procedure of the study was in accordance with the Declaration of Helsinki and was registered in ClinicalTrials.gov PRS (NCT02984553).

### 2.2. Data Aquisition

Triaxial accelerometers (Actigraph WGT3X-BT, Pensacola, FL, USA) placed over the right iliac crest in the mid-axillary line collected data from participants throughout the exercise tests. The device measured triaxial acceleration in the x, y, and z axis at a sampling frequency of 50 Hz for most of the subjects. The x-axis corresponds to anterior/posterior movement, y-axis corresponds to vertical movement, and the z-axis corresponds to medial/lateral movement. 39 of the 110 tests were recorded with a sampling frequency of 100 Hz; these were down sampled to 50 Hz prior to further analysis. ActiLife software (Version 6.8.1) was used to construct date and time stamped files of raw acceleration signal in the vertical, medial–lateral, and anterior–posterior planes.

The accelerometer data were obtained from two incremental speed tests performed on a treadmill ergometer (Kuntaväline, Hyper Treadmill 2040, Finland) and a 20-metre track marked with two cones; a detailed description of the trials and data collection protocols have been published previously [20,21]. The data set used for analysis combined 80 track tests and 40 treadmill tests that were completed in a single session and performed over 15 incremental stages, summarized in Table 1. Three minutes of standing data were collected in Stage 1, and 1.5 min of accelerometry data per stage were collected for the remaining stages. Each participant was free to start jogging from the 7th stage onwards (6.1 km/h), or the operator suggested to do so when the participant was not able to match the required speed.

### 2.3. Class Definitions

The combination of direct observation during the tests and offline visual inspection of raw accelerometry traces were used to define the two classification sets used within this study. The binary classes were ‘Standing’ and ‘Walking’ and the tertiary classes were ‘Standing’, ‘Walking’ and ‘Jogging’. These characteristics showed that across all participants and tests, the participants were ‘Standing’ at Stage 1 and ‘Walking’ between Stages 2–8. This formed the criteria for the binary classifier, which was the same across the whole data set. For the more complex tertiary classifier, the classes were defined on a test-by-test basis.

### 2.4. Data Pre-Processing

Prior to extracting features from the accelerometer signals, it was necessary to pre-process the data, including filtering out noise and segmentation. Typically, accelerometer signals may contain noise due to additional frequency bands related to acceleration due to gravity between 0–0.8 Hz; a high pass filter can be used to eliminate this noise, typically previously applied with a cut-off frequency (*f_c_*) at 0.5 Hz [22,23].

Windowing of accelerometer signals is extremely common among activity classification studies as a method of dividing the signal into various smaller segments [24]. This aids to increase the number of observations available per classification, thus increasing the number of ways each class may be described as features are extracted on a segment-by-segment basis. According to Banos et al. [24], the recommended window length for activity classification applications is between 1 and 2 s.

For this study, a custom Hamming window Finite Impulse Response (FIR) high pass filter with cut-off frequency *f_c_* = 0.5 Hz and an order of 1000 was designed using MATLAB’s Filter Designer App.

The accelerometer signals were split into 2-s windows, with 100 samples per window at the sampling frequency of the data, 50 Hz, like in a study by Preece et al. [25]. Additionally, a 50% overlap of the windows was used to double the segment size, as was proven useful in a study by Bao & Intille [26] to further increase observation numbers per category.

### 2.5. Feature Extraction

For each participant, various feature sets were extracted using different signal processing techniques, as shown in Table 2. A total of 18 features were extracted from each window of data. Out of these features, three: (Lempel Ziv (LZ) Complexity, Central Tendency Measure (CTM) and Correlation Dimension (D2)) have not been applied to the analysis of accelerometry data in human movement classification in previous studies. 

Identifier 1 is the arithmetic mean, computed for each time series window using Equation (1).
(1)$$\bar{x}=\sum \frac{x_{i}}{n},$$
where $\bar{x}$ is the mean, $\;x_{i\;}$ is the ith number in the series and n is the length of the series.

Identifier 2 is the median, computed as the middle value, n, of $\;x_{i}$, in each time series window using Equation (2).
(2)$$n=\frac{N+1}{2},$$
where n is the middle value and N is the length of the series.

Identifier 3 is the standard deviation, calculated using Equation (3).
(3)$$\sigma =\sqrt{\frac{1}{n-1}}\sum {\left(x_{i}-\bar{x}\right)}^{2},$$
where σ is the standard deviation, n is the length of the series, $x_{i}$ is the ith number in the series and $\bar{x}$ is the mean of the series.

Identifiers 4 and 5 are the 25th and 75th percentiles and are calculated as the values below which 25% or 27% of the data is found. 

Identifier 6 is skewness, which is a measure of the symmetry of the data, calculated using Equation (4).
(4)$$Skewness=m_{3}\sigma^{-\frac{3}{2}},$$
where $m_{3}\;$ is the third moment and σ is the standard deviation.

Identifier 7 is Kurtosis, which is a measure of the spread of a data distribution, calculated using Equation (5) [32].
(5)$$Kurtosis=m_{4}\sigma^{-2},$$
where $m_{4}$ is the fourth moment and σ is the standard deviation.

Identifier 8 is Principal Frequency, which is the frequency band that is computed to have the highest power in the signal. It is found by producing the frequency spectrum of a time series using the Fast Fourier Transform (FFT) and identifying the frequency with the highest power. The power can be estimated using Equation (6).
(6)$$Power=\frac{{\left|fft\left(x\right)\right|}^{2}}{n},$$
where fft is the Fast Fourier Transform of the time series, x  is the filtered time series and n is the length of the series.

Identifier 9 is Spectral Energy, defined as the sum of the squared FFT coefficients, calculated using Equation (7).
(7)$$Spectral\;Energy=\sum_{i=1}^{n}fft{\left(x_{i}\right)}^{2},$$
where fft  is the Fast Fourier Transform of the time series, x  is the filtered time series and n  is the length of the series.

Identifier 10 is the LZ Complexity which is a measure of the complexity of a signal and is a method of graining measurements. The signal is converted to a binary signal, where the numbers 0 and 1 describe whether the value is below or above a certain threshold, respectively. The threshold is commonly defined as the median of the sequence [33]. The coarse graining is calculated using Equation (8), and then is scanned from left to right using the Lempel-Ziv 1976 algorithm to identify the number of different sub-sequences in the time series [34]. That complexity count is normalised to obtain a LZ complexity value between 0 and 1.
(8)$$s\left(i\right)=\{\begin{array}{c} 0\;\;\;\;\;\;\;if\;x\left(i\right)<T_{d}\\ 1\;\;\;\;\;\;\;if\;x\left(i\right)\geq T_{d} \end{array},$$
where s(i)  is the coarse-grained sequency, x(i)  is the original series and $T_{d}\;$ is the threshold (median) of the series.

Identifier 11 corresponds to the CTM which is a measure of the variability encountered in a signal. A plot of the first differences of the signal provides a graphical representation of this variability of a time series. A radius, *ρ*, is defined, and the CTM is defined as the proportion of the data series which falls inside that region, calculated in Equation (9). For this study, *ρ* = 0.1 was chosen as it was able to inform a CTM value for both slower and faster speed stages.
(9)$$\begin{array}{c} CTM=\frac{\sum_{n=1}^{N-2}\delta \left(d_{n}\right)}{N-2},\\ \delta \left(d_{n}\right)=\{\begin{array}{c} 1,\;\;\;if{\left[{\left(x\left(n+2\right)-x\left(n+1\right)\right)}^{2}+{\left(x\left(n+1\right)-x\left(n\right)\right)}^{2}\right]}^{1/2}<\rho \;\\ 0,\;\;\;\;\;\;\;\;\;\;\;\;\;\;\;\;\;\;\;\;\;\;otherwise \end{array}, \end{array}$$
where N  is the length of the time series, ρ  is the radius and x  is the data series.

Identifier 12 is the D2 which is another measure of the dimensional complexity of a signal and is computed as described by Grassberger & Procaccia [35]. It is noted that meaningful results cannot traditionally be extracted from physiological data due to the large number of data required for D2, in addition to assuming the signal is stationary.

Identifiers 13 to 18 are calculated using the Discrete Wavelet Transform (DWT). In short, DWT passes the signal through a series of filter banks to decompose the signal into approximation (low-pass filtering) and detail (high-pass filtering) coefficients [36]. It produces the minimum number of coefficients required to recover the signal, which is ideal for efficiency computing compared to its counterpart, Continuous Wavelet Transform (CWT).

Using the features extracted from the data, nine different sets were created combining features of similar characteristics, shown in Table 3. Principle Component Analysis (PCA), used in Feature Sets B and D, was used to reduce the dimensionality of the feature sets. Neighbourhood Component Analysis (NCA) was used to select the features in sets C1 and C2. Any features with a weight of >0.2 were extracted and used in these sets.

NCA is a method that can be used to select the best features of a data set to avoid overfitting of machine learning models. It calculates the weights of each feature that are computed and is regularized with a parameter called Lambda (*λ*) [37].

PCA is a method that transforms a ser of correlated variables into a new set of uncorrelated variables in principal components. It involves determining the covariance matrix of the data set and subsequently decomposing the matrix using eigenvalue decomposition into the eigenvector matric and eigenvalues [38].

### 2.6. Machine Learning Classification

Four different Machine Learning classifiers, shown in Table 4, were employed in this study to predict the class of each segment of data within the selected Feature Set.

The k-Nearest Neighbours (k-NN) classifier is one of the simplest and most straightforward machine learning algorithms to implement. This classifier is distance-based and produces a model directly based on the available training data [38]. The classifier calculates the distance, in the feature space, between a given test point and the points in the training set, producing the k-nearest points. The value of k-nearest points is defined in the training of the model and for this study was set at 10, matching a recent study by Ryu et al. [39]. The class that most of the k-nearest points belongs to results in the predicted class for the test point. There are two different distance measures that are commonly used, the Euclidian distance and the Manhattan distance. The Euclidian distance, commonly known as the straight-line distance, gives the shortest distance between two points whereas the Manhattan distance is the sum of the absolute differences of each coordinate axis. The Euclidian distance is calculated using Equation (10) and the Manhattan distance is calculated using Equation (11).
(10)$$dist=\sum_{i-1}^{n}\sqrt{{\left(q_{i}-p_{i}\right)}^{2}},$$
where $p_{i}$ and $q_{i}$ are *n*-dimensional points.
(11)$$dist=\sum_{i-1}^{n}\sqrt{|q_{i}-p_{i}},$$
where $p_{i}$ and $q_{i}$ are *n*-dimensional points.

Classification tree models, or Decision Trees (DT), are based on dividing the feature space into several rectangles and fitting a unique model within each one. This process can be represented through a tree, as the name of the classifier suggests [40]. At the top of the tree, the model considers all the features of the data set. When analysing a test point, its value for the first feature is considered as the start. A decision of which class the data point might belong to is made based on information from the training data, subsequently defining the following branch of the tree. This is repeated for each feature, until a final decision of the predicted class of the test point is made. This method is known as CART [41].

A Support Vector Machine (SVM) produces decision boundaries with the aim of optimising a margin between two classes to obtain the best separation of classes as possible. The goal of an SVM is to identify the ideal hyperplane decision boundary. The SVM algorithm places data points in a feature space where each of the classes are set as −1 and 1. This is done to simplify the calculation of a hyperplane decision boundary, which would exist at 0. This assumption is used in conjunction with a linear discriminant, shown in Equation (12), to find the margin and maximise it according to input patterns and weight vector [38]. This process can also be used for three-class classification by splitting the problem into one-versus-one classification cases and combining the results, which adds complexity to algorithm.
(12)$$y=\sum_{i=1}^{M}x_{i}w_{i}+b,$$
where M is the number of features, $x_{i}$ are the input variables, $w_{i}$ are the model weights, b is the bias and y is −1 or 1 for each of the classes.

As detailed in Duarte et al. [42], k-NN, DT and SVM classifiers have previously been used in similar applications using frequency domain features to produce accuracy results that are greater than 90% for human activity recognition. Therefore, these classifiers were chosen for further analysis within this study.

### 2.7. Training Methods & Model Evaluation

The models outlined in Table 4 were trained and tested using 3 alternative methods with the idea to initially establish the best performing models and then assess their repeatability to acquire the best model overall. The first method combined all the track and treadmill data into a single data set and randomly assigned 70% of the data to a training set and the remaining 30% of the data to a test set. Randomising and combining the data allows a better chance of avoiding overfitting due to the increased data variability. The 30% test set provides a large pool of data to evaluate the performance of the model on without compromising the training set.

The second method is k-fold cross-validation. The entire data set is randomly split into k folds; in the case of this study, *k* = 10. For k tests, one fold is used to test the model and the remaining 9 folds are used to train the model. It minimizes any effect of the order of the data and is a common method used to evaluate the overall performance of a model.

The final method is commonly referred to as ‘leave-one-out’ cross-validation. This method takes the data sources, *n* = 110, and performs training using *n* − 1 = 109 of these sources and tests on the remaining data source. It repeats this process *n* number of times, so that each source becomes the test set. The results from the tests given an indication of the repeatability of the model’s performance.

The performance of the model within this study primarily relates to the percentage accuracy of the classification of the test set. The accuracy is calculated as the number of correct predictions, compared to the predefined ground truth labels, as a percentage of the total number of predictions made. Confusion Matrices is the reporting metrics used to assess the performance of machine learning classifiers. Confusion Matrices show the relationship between the number of predictions for each class and the number of correct/incorrect results. Per class accuracies within confusion matrices can be useful in assessing the performance of a classifier to understand bias. Due to the large number of tests detailed in the Results section, the confusion matrices are shown for the top three performing models.

## 3. Results

### 3.1. Binary Machine Learning Classification

Initially, each feature set was tested against each model for the binary classification problem using the 70% train, 30% test method. The results, shown in Table 5, show that overall, the k-NN classifier Manhattan and Feature Set C1 produced the highest accuracy results of 99.70%. The top 3 mean accuracy results across all models were Feature Set A, C1 and E. These feature sets were therefore taken forward for cross-validation evaluation.

The 10-fold cross-validation results for Feature Sets A, C1 and E for the binary classification problem are shown in Table 6. Feature Set A is a high dimensionality set that includes all calculated features. Feature set C1 reduced upon the dimensionality of Feature Set A using NCA, requiring significantly more computing power than Feature Set A and E. Feature Set E contains basic statistics and is the simplest to calculate out of the three sets. In addition to this, the very similar 10-fold cross-validation accuracy results displayed between the three feature sets, Feature Set E was chosen to be taken forward for leave-one-out cross-validation testing.

The confusion matrices for the top 3 performing models, highlighted yellow in Table 6, are shown in Figure 1a–c, respectively.

The leave-one-out cross-validation testing of Feature Set E for the binary classification problem is shown in Table 7. Similarly, to the cross-validation training, the k-NN Manhattan model classifier produced the overall best mean accuracy results. One participant (Treadmill 25) produced significantly poorer accuracy results (87.43% across all four models) than the overall mean. Treadmill 25 was one of only 2 treadmill subjects that completed the entire test up to speed stage 15, indicating the data in higher speed stages negatively skews the model’s performance.

### 3.2. Tertiary Machine Learning Classification

Each feature set was created for the tertiary classifier using all available Stages for each subject. Each feature set was tested against each model for the tertiary classification problem using the 70% train, 30% test method. 

The results, shown in Table 8, show again that the k-NN Manhattan classifier and Feature Set C2 produced the highest accuracy results of 95.56%. The top 4 mean accuracy results across all models were Feature Set A, C1, C2 and H. As C1 and C2 only differ by the adjusted NCA parameters used, C2 was removed, leaving Feature Set A, C1 and H taken forward for cross-validation evaluation.

The 10-fold cross-validation results for Feature Sets A, C1 and H for the tertiary classification problem are shown in Table 9. Feature Set A was chosen to be taken forward for leave-one-out cross-validation testing as it provided the most consistent results across the 4 classifiers.

The confusion matrices for the top 3 performing models, highlighted in yellow in Table 9, are shown in Figure 2a–c, respectively.

The leave-one-out cross-validation testing of Feature Set A for the tertiary classification problem is shown in Table 10. The SVM classifier was excluded from the analysis due to the significantly longer processing time required per subject (>40 min) compared to the other classifiers. Similarly to the binary models tested using this method, the Manhattan k-NN classifier produced the overall best mean accuracy results. In addition, three participants (Track 62, Track 64, and Treadmill 25) produced significantly poorer accuracy results than the overall mean. In addition to Treadmill 25, Track 62 and Track 64 were two of only five track subjects that completed the entire test up to speed stage 15, further indicating the data in higher speed stages negatively skews the model’s performance.

## 4. Discussion

To the best of our knowledge, this is the first study to develop and test a wide variety of feature extraction methods applied to machine learning models for the automatic identification of weight-bearing physical activity types in postmenopausal women from hip-worn commercial devices. This study included a significant number of tests utilising various feature sets and classifiers. It also provided in depth analysis of these feature sets and classifiers, as well as the results from them. Furthermore, our classifiers were developed from two different sets of accelerometer data (track and treadmill) collected in the laboratory, which enabled the control of a range of activities and intensities which allowed calibration of the results. The hip is the most common wear location for Actigraph devices in longitudinal and intervention studies in the bone health field, so the accuracy of models for this location and device has relevance for researchers and health practitioners [43].

The highest overall cross-validated accuracy for the binary classifier was 99.63% using Feature Set E (Basic Statistics) and the k-NN Manhattan classifier. Almost identical accuracies were seen for both the k-NN Manhattan and k-NN Euclidian classifiers across Feature Sets A (All Features), C1 (NCA Selected Features, *λ* = default) and E. The tertiary classifier had a much wider spread between cross-validated accuracies, with 95.48% for Feature Set A using the k-NN Manhattan classifier to 90.43% for Feature Set H (Wavelets) using the DT classifier. The leave-one-out cross-validation results were slightly lower than the cross-validated results, with a mean of 99.61% for the binary k-NN Manhattan classifier using Feature Set E and 94.04% for the tertiary k-NN Manhattan classifier using Feature Set A. The use of the leave-one-subject-out cross-validation, ideal for smaller data sets, highlights any overfitting of the machine learning models and challenges the model’s general applicability to new data which is needed for clinical applications [44,45]. Therefore, the results obtained are expected as the classifiers perform sub-optimally for some individual subjects which could be attributed to overfitting within the model and the expectation that there may be lower performance when working with human experimental data. Interestingly, even though the k-NN Manhattan classifier was the simplest, it produced the highest performance, suggesting that the other classifiers may have been overcomplicating the classification task.

The data set used has a clear bias towards the walking class, as there was only one standing stage and less running speed stages per subject. This is highlighted in the binary results, where over the three highest accuracy models, the walking class had an average prediction error of 0.2% and the standing class had an average prediction error of 1.3%. This bias is shown in a different way for the tertiary classifier, where the average prediction error of the running class was 18.3% compared to 2.2% for the standing class and 2.9% for the walking class. As expected, this high percentage for the running class is made up of mostly incorrect walking predictions, possibly due to similar patterns in the feature sets between walking and running. This bias could also be attributed to instances where were participants felt uncomfortable to walk at higher speeds. Therefore, they started to combine walking and running for short periods to match the required speed for the test.

In terms of feature sets, it is also worth noting that the complex computational and feature reduction methods used in sets B (3D PCA Reduced), C1, C2 (NCA Selected Features, *λ* = 0), D (NCA Selected Features and 3D PCA Reduced) typically did not yield the best results. This could suggest that, in the case of Feature Set A, the large number of features (18) provided more information to the model and therefore yielded more accurate predictions. In contrast, in the case of Feature Set E, the features that carry the most important information for this classification are held in simple, well-known, and quick to calculate features. 

For real-time applications, it is vital that a machine learning model computes results quickly as well as accurately. Using the examples above, Feature Set E consisting of basic statistics is much less computationally expensive than Feature Set A which required complex non-linear calculations to be conducted. The computational time is affected by the dimensionality of the feature set, indicating that the optimum combination for real-time applications would be simple features with a low dimensionality. 

Although comparing results between studies is complicated due to differences in populations, tests and monitor placements that may influence classification, our overall finding agrees with previous studies showing that machine learning algorithms can be used to produce accuracies greater than 80% [46]. Similar research that conducted laboratory-based testing to validate types of activity algorithms including machine learning has provided per class accuracies greater than 99% for standing, walking, and jogging [47]. However, these studies included fewer subjects and samples and a monitor placement that could provide more obvious detectable changes between different activities, compared to our study. Additionally, Pires et al. [48] provided a detailed review of many additional studies within activity recognition using accelerometry data and machine learning, further suggesting that these methods can produce accurate and repeatable results [48]. In addition, our results further improve on many published papers within this field as different verification methods are analysed within our study to describe the robustness of the proposed method.

Results from our study support the use of accelerometers and machine learning approaches to classify physical activity types. Accurate recognition of stationary behaviour [49] from weight-bearing physical activity was achieved using features consisting of basic statistics with a low dimensionality. This finding is of significance in the context of increased emphasis on changes in habitual physical activity and for future understanding of the long-term health implications of stationary behaviour [50]. Our classifiers were able to accurately detect bouts of walking and jogging, which are important activities for several health outcomes [51] including skeletal health. Clinical trials have shown that walking improves femoral neck bone mineral density [52], and population studies have found that participation in walking since age 50 is associated with higher levels of osteogenic physical activities in older age [53]. Similarly, an analysis of UK Biobank data found that accumulating 1–2 min/day or high intensity PA equivalent to slow jogging in postmenopausal women was associated with better bone health [54]. Automatic recognition of walking and jogging is therefore critical to monitor the effectiveness of interventions and may help clinicians and policy makers with public health messaging. The accurate classification of these specific activities may also lead to fully understanding relationships between duration and intensity of weight-bearing PA and different health outcomes in large-scale studies that typically use commercially available activity monitors like the Actigraph; ultimately making clinical recommendations easier. 

Nevertheless, our study also has several limitations that should be acknowledged. The models were trained on a relatively small number of healthy postmenopausal and may not reflect the full range of movement patterns and intensity associated with activity in all post-menopausal women. Moreover, the machine learning algorithms were developed and tested on activity performed under controlled laboratory conditions, which may limit their performance during free-living activities [55]. However, the laboratory-based data set did contain walking and jogging activities that were performed with varying linear (treadmill) and dynamic turns (track). The classifiers developed in this study should be validated in a free-living set-up and compared to a similar algorithm developed on free-living data.

## 5. Conclusions

This study presents a method to produce highly accurate results for the classification of standing, walking and jogging of postmenopausal women using triaxial accelerometry data. The process involved pre-processing and extracting 18 different features, from the raw data, which were then grouped into 9 unique feature sets. These feature sets were used as the input to four different machine learning models, k-NN Manhattan, k-NN Euclidian, Decision Tree & SVM, for a binary (standing and walking) and tertiary (standing, walking and running) classification problems. The results from the models were initially tested using 70% of the data for training and the remaining 30% for testing. From these results, the top 3 highest accuracy models were then validated using two different methods: 10-fold cross-validation and leave-one-out validation. For the binary classifier, the best performing classifier used Feature Set E, which consisted of basic statistical features, as an input to a k-NN Manhattan classifier. This combination yielded the highest overall cross-validated accuracy of 99.63%. For the tertiary classifier, the best performing classifier used Feature Set A, which consisted of a combination of all the 18 features from linear and non-linear methods, as an input to a k-NN Manhattan classifier. This combination yielded the highest overall cross-validated accuracy of 95.48%. The leave-one out results from these same models yielded binary classification accuracy results of 99.61% and tertiary classification accuracy results of 94.04%. The results of this test indicated that these models are, on average, accurate when applied to individual data sets and suggests that the models exhibit minimal overfitting. Low overfitting is a desired trait when applying machine learning models to unseen data. Overall, these models have produced high accuracy results with more rigorous validation testing when compared to similar studies within this field.

Our results further suggest that future studies could use these classifiers to characterise weight-bearing PA, such as walking and jogging, more accurately and provide a basis for PA based policies and interventions intended to improve bone health and a broad range of health outcomes in this age group. Alternative data sets could also be applied to these models to further enhance the classifiers’ use case. These data sets could include formats such as free-living accelerometry data or accelerometry data sets from cohort studies. For example, the classifier could be used to analyse raw accelerometry datasets from cohort studies which would allow the relationships between specific types of physical activity and important health outcomes to be investigated further. Accurate classifiers on free-living accelerometry data would advance these tools towards usability in a clinical setting. Their properties could include longer assessment times, populations with multiple long-term conditions, a wider range of recorded activities and additional or alternatively located accelerometers; all of which would provide additional information for wider classification decisions.

## Figures and Tables

**Figure 1 sensors-22-09176-f001:**
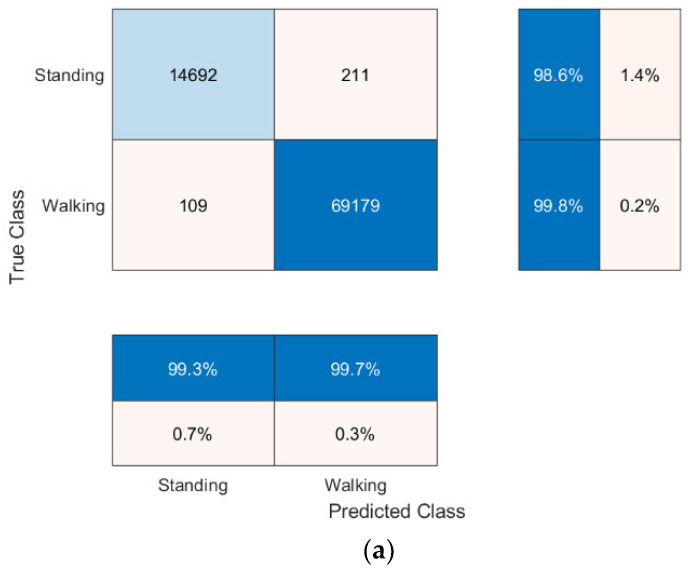
Confusion matrices for (**a**) Binary Classifier, Feature Set A, Model.1 (k-NN Manhattan Classifier). (**b**) Binary Classifier, Feature Set E, Model.1 (k-NN Manhattan Classifier). (**c**) Binary Classifier, Feature Set E, Model.2 (k-NN Euclidian Classifier).

**Figure 2 sensors-22-09176-f002:**
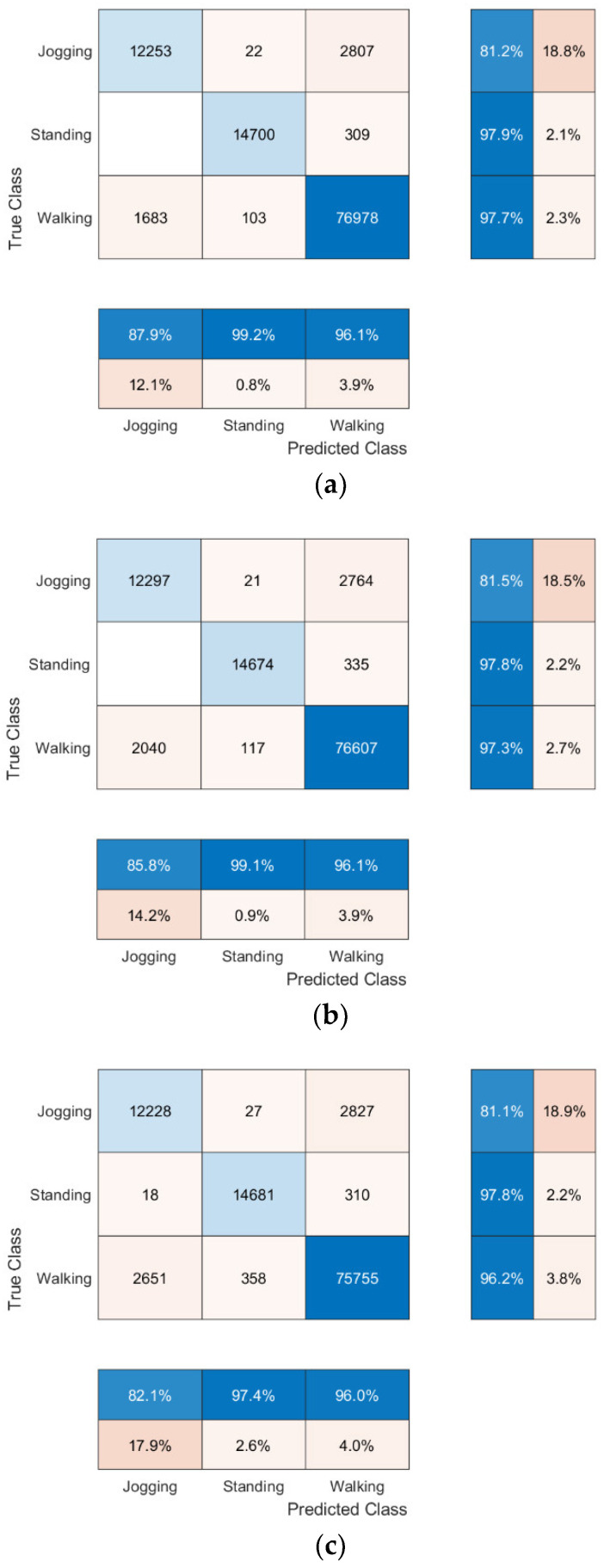
Confusion matrices for (**a**) Tertiary Classifier, Feature Set A, Model.1 (k-NN Manhattan Classifier). (**b**) Tertiary Classifier, Feature Set C1, Model.1 (k-NN Manhattan Classifier). (**c**) Tertiary Classifier, Feature Set A, Model.3 (Decision Tree Classifier).

**Table 1 sensors-22-09176-t001:** The stage number and corresponding speed of movement for the track and treadmill tests.

Stage	1	2	3	4	5	6	7	8	9	10	11	12	13	14	15
**Speed (km/h)**	0	2.4	3.0	3.6	4.3	4.9	5.5.	6.1	6.7	7.3	7.9	8.5	9.1	9.8	10.4

**Table 2 sensors-22-09176-t002:** List of features extracted from the pre-processed data set, traced throughout this report using a unique identifier and linked to research papers each feature is used within.

Unique Identifier	Feature	Papers Used within
1	Mean	[17,18,25,26,27,28,29,30]
2	Median	[25]
3	Standard Deviation	[18,25,28]
4, 5	25th and 75th Percentiles	[25,28,30]
6	Skewness	[18,27,28]
7	Kurtosis	[18,27,28]
8	Principal Frequency	[22,23,25,27]
9	Spectral Energy	[17,25,26,28,29,31]
10	LZ Complexity	N/A
11	CTM	N/A
12	D2	N/A
13	Wavelet 1, Level 1 (Daubechies 2 Mother Wavelet)	[25]
14	Wavelet 2, Level 1 (Daubechies 2 Mother Wavelet)	[25]
15–16	Wavelet 1, Levels 2–3 (Daubechies 2 Mother Wavelet)	[25]
17–18	Wavelet 2, Levels 2–3 (Daubechies 2 Mother Wavelet)	[25]

**Table 3 sensors-22-09176-t003:** List of feature set identifiers and names, with a description of each and the dimensionality of each feature set.

Unique Identifier	Feature Set Name	Description	Dimensionality
A	All Features	All features described in Table 2	54
B	3D PCA Reduced	PCA Reduction of Feature Set A	3
C1	NCA Selected Features	NCA Selection of Features, *λ* = default	Model Dependent
C2	NCA Selected Features	NCA Selection of Features, *λ* = 0	Model Dependent
D	NCA Selected Features and 3D PCA Reduced	PCA Reduction of Feature Set C2	3
E	Basic Statistics	Features 1–5	15
F	Frequency Domain	Features 8–9	6
G	Non-Linear Features	Features 10–12	9
H	Wavelets	Features 13–18	18

**Table 4 sensors-22-09176-t004:** A list of machine learning models used and their associated descriptions.

Model (X = Binary or Tertiary)	Description
X.1	k-NN Classifier (Manhattan)
X.2	k-NN Classifier (Euclidian)
X.3	Decision Tree (DT) Classifier
X.4	Support Vector Machine (SVM)

**Table 5 sensors-22-09176-t005:** A summary of each feature set paired with each model for the binary classification problem, tested using the 70% train, 30% test method. The mean accuracy value for each feature set and model number are also shown.

		Total No. Data Points	Model Number				
			Binary.1	Binary.2	Binary.3	Binary.4	MEAN
**Feature Set**	**A**	84,191	99.62	99.61	99.38	99.56	99.54
	**B**	84,191	96.31	96.25	94.41	90.93	94.48
	**C1**	84,191	99.70	99.68	99.37	99.63	99.60
	**C2**	84,191	99.58	99.57	99.33	99.55	99.51
	**D**	84,191	96.37	96.36	94.75	94.80	95.57
	**E**	88,892	99.59	99.57	99.31	99.53	99.50
	**F**	88,892	98.04	97.87	99.42	95.55	97.72
	**G**	84,191	99.32	99.19	99.10	99.41	99.26
	**H**	88,892	99.55	99.53	99.31	99.54	99.48
**MEAN**			98.68	98.63	98.26	97.61	

**Table 6 sensors-22-09176-t006:** A summary of Feature Sets A, E and H pair with each model for the binary classification problem, tested using 10-fold cross-validation. The mean percentage accuracy values for each feature set and model number are also shown. The results highlighted in yellow show the top 3 performing models in terms of mean percentage accuracy.

		Total No. Data Points	Model Number				
			Binary.1	Binary.2	Binary.3	Binary.4	MEAN
**Feature Set**	**A**	84,191	99.62	99.62	99.34	99.60	99.55
	**C1**	84,191	99.60	99.60	99.32	99.57	99.52
	**E**	84,191	99.63	99.62	99.37	99.60	99.56
**MEAN**			99.62	99.61	99.34	99.59	

**Table 7 sensors-22-09176-t007:** A summary of Feature Set E with each model for the binary classification problem, tested using leave-one-out cross-validation. The mean accuracy, standard deviation (SD), minimum and maximum values for each model number are also shown.

Feature Set E	Model Number			
	Binary.1	Binary.2	Binary.3	Binary.4
**Mean**	99.61	99.61	99.43	99.58
**SD**	1.34	1.34	1.34	1.36
**Min**	87.43	87.43	87.43	87.43
**Max**	100.00	100.00	100.00	100.00

**Table 8 sensors-22-09176-t008:** A summary of each feature set paired with each model for the binary classification problem, tested using the 70% train, 30% test method. The mean accuracy value for each feature set and model number are also shown.

		Total No. Data Points	Model Number				
			Tertiary.1	Tertiary.2	Tertiary.3	Tertiary.4	MEAN
**Feature Set**	**A**	108,855	95.47	93.95	94.17	92.46	94.53
	**B**	108,855	84.71	84.80	81.28	72.93	83.60
	**C1**	108,855	94.85	93.78	94.12	91.89	94.25
	**C2**	108,855	95.56	93.90	94.13	92.35	94.53
	**D**	108,855	85.67	85.70	81.80	75.40	84.39
	**E**	113,450	91.49	91.36	87.84	89.58	90.23
	**F**	113,450	91.23	90.83	92.55	81.38	91.54
	**G**	108,855	88.66	88.16	86.51	87.86	87.78
	**H**	113,450	92.91	92.81	90.52	91.29	92.08
**MEAN**			91.17	90.59	89.21	86.13	

**Table 9 sensors-22-09176-t009:** A summary of Feature Sets A, E and H paired with each model for the tertiary classification problem, tested using 10-fold cross-validation. The mean percentage accuracy values for each feature set and model number are also shown. The results highlighted in yellow show the top 3 performing models in terms of mean percentage accuracy.

		Total No. Data Points	Model Number				
			Tertiary.1	Tertiary.2	Tertiary.3	Tertiary.4	MEAN
**Feature Set**	**A**	84,191	95.48	94.02	94.31	92.30	94.03
	**C1**	84,191	95.15	93.83	94.26	91.91	93.79
	**E**	84,191	93.06	92.98	90.43	91.44	91.98
**MEAN**			94.56	93.61	93.00	91.88	

**Table 10 sensors-22-09176-t010:** A summary of Feature Set A with each model for the tertiary classification problem, tested using leave-one-out cross-validation. The mean, standard deviation (SD), minimum and maximum values for each model number are also shown.

Feature Set E	Model Number			
	Tertiary.1	Tertiary.2	Tertiary.3	Tertiary.4
**Mean**	94.04	92.96	93.14	N/A
**SD**	5.33	5.39	5.19	N/A
**Min**	69.82	68.54	72.09	N/A
**Max**	100.00	100.00	99.45	N/A

## Data Availability

Data is available upon reasonable request to the corresponding author.

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
