# Peer review of "Machine Learning Models for Weight-Bearing Activity Type Recognition Based on Accelerometry in Postmenopausal Women"

_sensors, 2022, doi:10.3390/s22239176_

Round 1

Reviewer 1 Report

General comments

The article Machine learning models for weight-bearing activity type recognition based on accelerometry in postmenopausal women is well written and well organized. The research topic is original and addresses the important issue of evaluating the accuracy of four machine learning models on binary (standing and walking) and tertiary (standing, walking and jogging) classification tasks in postmenopausal women. The methodology adopted in the paper is described succinctly and clearly, but in my opinion it needs a few minor additions.  The results have been discussed in a  correct manner allowing for easy interpretation.  Minor corrections will be necessary before the work is accepted for publication.

Minor comments:

Introduction

The introduction should be expanded. The research problem should be emphasized more based on the existing literature. The number of works cited in the introduction currently amounts to only 12 items, most of which are works older than 5 years. Please expand the introduction based on the latest literature and add relevant references.

Materials and Methods

Subsection 2.5 Feature Extraction should be gently restructured and supplemented with brief information on each indicator with the necessary formulas. Authors can find useful information in the works:

https://doi.org/10.1016/j.asoc.2015.02.015

https://doi.org/10.3390/s22103765

Jedliński, J. Caban, L. Krzywonos, S. Wierzbicki, and F. Brumerčík, “Application of vibration signal in the diagnosis of IC engine valve clearance,” Journal of Vibroengineering, Vol. 17, No. 1, pp. 175–187, Feb. 2015

Results and Discussion

The chapter is written correctly and the results are presented in a reproducible manner that allows for comparison. The results have been cross-referenced with those obtained by other authors.

Conclusion

The conclusions are supported by the results obtained were presented correctly in a concise manner.

After making the appropriate additions, the article may be accepted for publication.

Author Response

Please see attachment. Reviewer 1 & 2 comments left together for completeness.

Reviewer 2 Report

The manuscript is well structured and methodologically robust. It addresses a very interesting research niche with strong practical implications. However, the theoretical component, especially at the machine learning level, needs to be improved.

Improvement suggestions:

- From line 47 to 61, the authors use the expression “however” three times. Please try to avoid it. There are different ways to write it.

- Citation model in this sentence is incorrect: “For example, Freedson et al. (2011)….” A similar issue appears in “In a more recent study, Hagenbuchner et al. (2015) applied machine learning approaches…”

- Authors note in the introduction section “The purpose of this study was to evaluate four machine learning models (k-Nearest Neighbours (NN) Manhattan, k-NN Euclidian, Decision Tree (DT) and Support Vector Machine (SVM))” What are the reasons for choosing these models?

- More detailed information regarding the theory that are based these adopted models should be given. There is space to improve the theoretical background of this work.

- I realized that the authors provide a very brief overview of the four machine learning models adopted in this study. This description lacks more theoretical depth. There is space to include more relevant references.

- It is not clear where the data was collected. In which country or countries did this study take place?

- The theoretical and practical contributions of this work should be clearer in the Conclusions section.

- More exploratory theoretical work on machine learning models is needed. The references used in this area are insufficient.

Author Response

Please see the attachment. Reviewer 1 & 2 comments left together for completeness.

Round 2

Reviewer 2 Report

I recommend the authors to implement a minor improvement. They suggested “Alternative data sets could also be applied to these models to further enhance the classifiers’ use case.” It would be interest to provide a good vision about what might be a good data set. Which proprieties it should have?
